# Corn Cob as a Green Support for Laccase Immobilization—Application on Decolorization of Remazol Brilliant Blue R

**DOI:** 10.3390/ijms23169363

**Published:** 2022-08-19

**Authors:** Priscila M. dos Santos, Julia R. Baruque, Regiane K. de Souza Lira, Selma G. F. Leite, Rodrigo P. do Nascimento, Cristiano P. Borges, Robert Wojcieszak, Ivaldo Itabaiana

**Affiliations:** 1Department of Biochemical Engineering, School of Chemistry, Federal University of Rio de Janeiro, Rio de Janeiro 21941-909, Brazil; 2COPPE/Chemical Engineering Program, Federal University of Rio de Janeiro, Rio de Janeiro 21941-972, Brazil; 3CNRS, Centrale Lille, UMR 8181—UCCS—Unité de Catalyse et Chimie du Solide, University Lille, University Artois, F-59000 Lille, France

**Keywords:** laccase, laccase immobilization, corn cob, lignocellulosic biomass, dye degradation

## Abstract

The high demand for food and energy imposed by the increased life expectancy of the population has driven agricultural activity, which is reflected in the larger quantities of agro-industrial waste generated, and requires new forms of use. Brazil has the greatest biodiversity in the world, where corn is one of the main agricultural genres, and where over 40% of the waste generated is from cobs without an efficient destination. With the aim of the valorization of these residues, we proposed to study the immobilization of laccase from *Aspergillus* spp. (L_Asp_) in residual corn cob and its application in the degradation of Remazol Brilliant Blue R (RBBR) dye. The highest yields in immobilized protein (75%) and residual activity (40%) were obtained at pH 7.0 and an enzyme concentration of 0.1 g.mL^−1^, whose expressed enzyme activity was 1854 U.kg^−1^. At a temperature of 60 °C, more than 90% of the initial activity present in the immobilized biocatalyst was maintained. The immobilized enzyme showed higher efficiency in the degradation (64%) of RBBR dye in 48 h, with improvement in the process in 72 h (75%). The new biocatalyst showed operational efficiency during three cycles, and a higher degradation rate than the free enzyme, making it a competitive biocatalyst and amenable to industrial applications.

## 1. Introduction

Due to intense development in several strategic sectors, the world population has grown alarmingly in recent decades, which arouses intense concern about meeting the increasingly high demands for food and energy—where the increase in agricultural activity and consumer goods as development strategies is notable [1,2]. As a consequence, inestimable amounts of agro-industrial residues have been generated, and the conception of new strategies for their better management and valorization are urgent [3]. Moreover, human dependence on fossil fuels requires the development of new sustainable protocols for the use of renewable sources to meet future energy and chemical needs as a way to reduce emissions of greenhouse gases, which cause climate change and global warming, and severely threaten the current environment [4,5,6]. Lignocellulosic biomass (LB) is the largest source of renewable energy, composed of cellulose (40–50%), hemicellulose (15–25%), and lignin (15–30%) [7,8,9]. Due to its complex chemistry, LB has been used to obtain bio-based fuels or value-added chemicals [10,11]. However, the increasing amounts of agro-industrial residues generated worldwide are still a great biotechnological window of opportunity. In this trend, the integration between technologies developed by researchers and industry has been crucial to the search for short-term solutions, where the reuse of waste and the search for new (bio)catalysts are essential for obtaining more selective processes with higher yields in biorefineries in the future [12,13,14,15].

Brazil has some of the greatest biodiversity of the planet, containing about 70% of all the world’s flora and fauna. As a consequence, Brazilian agricultural activity is one of the most significant in the world, being internationally recognized for the production of maize (*Zea mays* L.), occupying third place in area and volume of production—preceded only by the United States and China [16,17]. The cob is a major byproduct of the corn industry, where it is obtained during the harvest and, in most cases, does not have a defined use, thereby becoming a residue [18]. It is estimated that in Brazil for each ton of processed corn, approximately 180 kg of corn cob is generated, where about 25% of the mass of these residues are properly reused [19,20]. In this context, a better valorization of this residual LB can be an auspicious source of new added-value products, such as new matrices for enzyme immobilization [21]. 

A large number of synthetic dyes have been intensively applied by several industries, such as pharmaceutical, fine chemistry, textile, and others [22,23]. Most of these compounds, due to their complex chemical nature, are difficult to degrade, in addition to their high toxicity, mutagenicity, and high risk of accumulation in soils, groundwater, and throughout the local ecosystem, which can cause environmental problems if not properly managed [24,25]. Several methods of treatment of effluents containing synthetic dyes have been developed, including physical methods of adsorption, complexation, and coagulation; in addition to those chemicals, oxidoreduction and hydrolysis are the best explored [26,27,28,29,30]. However, these current protocols present high costs and low selectivity, as well as lengthy operation times and uncertainties regarding the toxicity of the derivatives obtained [31]. For this reason, the biodegradation of dyes has received considerable attention in recent years, where detoxification catalyzed by laccases has been extensively explored in large-scale processes. 

Laccases (E.C.1.10.3.2, *p*-diphenol: dioxygen oxidoreductases), are multicopper-coordinated oxidases that are able to catalyze the oxidation of several phenolic (aminophenols, alkoxyphenols, catechols, quinols, and others) and non-phenolic compounds due to the catalytic reduction of molecular oxygen to water, and they can accept a wide range of substrates that can be further extended through combination with mediators [32,33,34]. In nature, laccases are present in plants, bacteria, insects, and white rot fungi [35,36]. Due to their low specificities, laccases have been applied in several processes such as biomass delignification [34,37,38], decolorization [39], bleaching of denim and paper [40], and the elimination of phenolics [41], among other things. However, its application in aqueous environments for the detoxification of effluents is limited due to its high thermal instability, restricted pH optimum of activity, as well as the ease of inhibition by the product and its non-reusability [42,43,44].

The immobilization of laccases on solid and insoluble supports can overcome these challenges and lead to a more sustainable approach, since it allows the reuse of the enzyme, and can improve the operational stability and promote the improvement and retention of catalytic activity, which can reduce the cost in several processes and increase the possibilities of developing new applications—since immobilized enzymes can be applied in different types of reactions [45,46,47]. Currently, several protocols have reported the immobilization of laccases from various sources over a range of synthetic, commercial supports, which ends up increasing the final costs of the process to be implemented [47,48]. In addition, several methodologies involve steps for the functionalization and derivatization of the supports in order to promote covalent bonds with the enzyme. Although immobilization by covalent bonding is a technique that promotes the formation of a robust biocatalyst, the several steps involved make the process costly and more difficult to reproduce [46]. Thus, the search for cheaper and more sustainable supports, as well as more simplified immobilization methods, are important aspects when choosing the construction of a biocatalyst. In this scenario, the hydrophobic adsorption methodology becomes an attractive technique, mainly because of its low cost, simplicity, and given that it does not require previous modifications of the support and can generate high rates of enzyme association with the support with high stability, relative activity, and recyclability [49,50,51]. In addition, due to its varied morphologies and appreciable hydrophobicity and rigidity, LB can also be applied as a support for the immobilization of enzymes, making it a renewable, abundant, and low-cost support for obtaining new biocatalysts to be applied in the enhancement of several processes [52,53], which is one of our aims in this work. As a continuation of our efforts for the valorization of Brazilian residual biomass [54], in this work we proposed the study of the immobilization of laccase from *Aspergillus* spp. in residual corn cob by an adsorption protocol, with a subsequent evaluation of the effect of the main variables (pH, amount of enzyme, time, recovered activity) on the activity of the final biocatalyst. In a second step, the new biocatalyst was applied in the degradation of the dye, Remazol Brilliant Blue R (RBBR), with the aim of determining its operational stability. So far, no work in the literature has reported the immobilization of laccases by hydrophobic adsorption on residual corn cob as well as its application on RBBR degradation, which motivated us to develop this work.

## 2. Results and Discussion

### 2.1. Immobilization of L_Asp_

The main objective of this work was to valorize the residual biomass of corn cob as a support for the immobilization of laccase from *Aspergillus* spp. (L_Asp_) in order to establish a simple protocol without onerous steps of functionalization and derivatization for the development of robust biocatalysts with operational stability and retention of enzymatic activity, and with subsequent application in the detoxification of the textile dye RBBR. For this reason, as a continuation of the previous work of our research group [55], the residual corn cob (CC) that was previously defatted, ground, comminuted, characterized, and selected for this study, which began with the investigation of the initial amount of protein offered to the defatted residual support (Table 1), followed the immobilization process by hydrophobic adsorption at pH 7.0 (phosphate buffer 0.1) and had an immobilization time of 24 h.

As can be observed in Figure 1, the activity expressed by the final immobilized biocatalyst showed an increase according to the increasing protein load added. However, this relationship was not linear, where the highest activities were found in biocatalysts containing 0.1 g.mL^−1^, whose expressed activity was 1854 U.kg^−1^ (Table 1). In higher concentrations of protein, the activities of the final biocatalyst showed a significant decrease. This effect may be related to a possible saturation of the support at higher concentrations of protein, where the higher loads may cause a tumultuous occupation of the support, thereby causing less access to the substrate and subsequently influencing the activity values obtained [46,47].

In order to observe the effect of pH on improving immobilization yields and recovered activity, L_Asp_ was immobilized, while maintaining a concentration of 0.1 mg.mL^−1^ and in a pH ranging from 4 to 9, as shown in Figure 2.

As observed, at the lowest pHs investigated, the lowest immobilization yields were found. At pH 7.0, in fact, the highest yields in immobilized protein (75%) and residual activity (40%) were obtained (Data available on Appendix A). Probably, at this pH, the L_Asp_ is in a more hydrophobic state with a considerable proportion of amino acids in their non-ionized form, thus providing greater interaction with the support [40,41]. 

In order to improve the immobilization time and find the best conditions for the immobilization yield and recovered activity, we studied the immobilization of L_Asp_ in CC over time, keeping the initial concentration of enzymes at 0.1 g.mL^−1^ immobilization and pH 7.0, whose results are shown in Figure 3.

As evidenced, increasing values of immobilization efficiency of L_Asp_ were obtained during times from 0.5 to 2 h, suffering negligible oscillations until 24 h of process, where 75% immobilization yield values remained stable, which makes the production of new biocatalyst more economically viable—since there was a reduction of around 12 times for the maximum protein loading, thereby demonstrating that the point of saturation of the support happens during this time (2 h). When observing the values of recovered activity found, a similar tendency of growth is found, where at 2.5 h of immobilization, 45% of recovered activity was obtained, demonstrating that the proteins immobilized in the support were in fact available to express their activity, and thereby corroborating with the data obtained in Figure 1—where the load of 0.1 g.mL^−1^ was demonstrated to be the best initial amount of protein to be offered to the system in order to avoid the tumultuous immobilization of protein in the support (all data available on Appendix A). 

The adsorption of an enzyme on a support is a dynamic process, being the result of multiple factors, where the initial protein load, the ionic strength, the surface and area of the support, its geometry, and the immobilization time play crucial roles in the success of the technique [56]. In hydrophobic adsorption processes, the hydrophobicity and porosity of the support are very important, because they are directly related to the surface area available for enzyme deposition and can generate different types of interactions, both hydrophobic interactions and those of accommodation and reticulation, which favor both the adsorption of larger quantities of proteins, as well as their better distribution by the support—besides greater protection against deleterious actions [57,58]. The adsorption of enzymes is not restricted to a single layer arranged by the support. Thus, when an excess of proteins is initially offered to the system, new layers of proteins may be formed on the support, causing a reduction of the activity recovered from the biocatalyst, since a large number of adsorbed proteins are not available to exert activity. In this study, an immobilization time of up to 2 h, which probably offers the best arrangement of proteins by the support, was found at the concentration of 0.1 g.mL^−1^, where it had previously been demonstrated that for higher concentrations, the values of recovered activity decreased—probably because of the formation of other layers of proteins by the support. Our values of immobilization yield (75.5%) and recovered activity (40%) were higher than the ones found by Coelho et al. (2011) [54], who realized the immobilization of the laccase Novozym 51003 (commercial enzyme with mediator) by the adsorption methodology in coconut fiber residues, where it was demonstrated that at concentrations of 0.067 g.mL^−1^, the highest values of recovered activity (12%) and immobilization efficiency (38%) were found; as well, they also had considerable reductions in these values when larger protein loads were offered, thereby demonstrating the tumultuous immobilization of proteins. After characterization of the support, it was demonstrated that it presented a smooth surface, where the immobilization at high concentrations of proteins happened in regular layers; thus, in lower concentrations of proteins, a first layer of enzymes was regularly deposited on high, and a second layer of enzymes was adsorbed on the first layer of enzymes, decreasing the efficiency of the system. These results become even more interesting since, besides the degreasing step, no other pre-treatment that aimed to deconstruct the biomass was performed. Several lignocellulosic supports have been applied in the immobilization of laccases. However, a large part of these ends up performing extensive pre-treatment protocols, making the final biocatalyst more expensive [51,59,60,61,62,63].

In order to better characterize the obtained biocatalyst, as well as optimize the immobilization protocol, the influence of ionic strength on the immobilization of L_Asp_ was also investigated (Figure 4). 

As the results indicate, the best immobilization yields were found when the enzyme was incubated at the lowest ionic strengths. In these conditions, the enzymes are normally solvated with the lowest ionic charge possible, increasing the possibilities of hydrophobic interactions with the support when the enzyme is conditioned in a pH where a considerable percentage of its amino acids are in the non-ionized form. According to the literature, the immobilization of laccases by adsorption have been studied in pHs ranging between 5.0 and 7.0. However, there are few papers where the ionic strength of the phosphate buffer is truly investigated. Thus, under conditions of lower ionic strength, increasing concentrations of laccase could be adsorbed on the support under conditions of ionic strength of 0.01 and 0.025 mol.L^−1^ (both approximately 83%—Data table available on Appendix A), with subtle reductions at higher concentrations, which are likely because there is less possibility of hydrophobic interactions with the support, since the enzyme is solvated by larger ionic layers at these concentrations. However, at these concentrations, the activities recovered from the biocatalysts were also low, being 11.5% at an ionic strength of 0.01 mol.L^−1^ and 23.8% at an ionic strength of 0.025 mol.L^−1^. As there were the highest possible concentrations of proteins in the supports, the multiple interactions may have formed several layers of proteins, and not making all the enzyme molecules available to exert their activity, or the enzyme molecules were adsorbed in a non-active conformation. Since it is also a porous support, under these conditions, a large load of enzyme can be lodged in the pores of the support, thereby causing difficulties in the diffusion of substrates and products, since access to the enzyme is probably hindered. In the condition of an ionic strength of 0.05 mol.L^−1^, although it was found that there was a slightly lower immobilization yield than in the lower concentrations (76%), the highest recovered activity was obtained (50.7%), demonstrating that at this ionic strength, the interactions made possible between enzyme and support provided the most favorable conditions for L_Asp_ to exert its activity, which is probably accomplished by providing the immobilization of a concentration of proteins sufficient to fill the support more regularly than when in conditions of lower ionic strength [59,64]. In conditions of higher ionic strength, lower recovered activities were also observed. Probably, in these conditions, the enzyme molecules may have been immobilized preferentially by lodging in the pores of the support, since in solvation by larger amounts of charge, the enzyme presents less possibilities of hydrophobic adsorption to the surface. Therefore, the condition of an ionic strength of 0.05 mol.L^−1^ was chosen for subsequent studies. Taking into account that 76% of the immobilization yield was obtained at an ionic strength of 0.05 mol.L^−1^, it can be inferred, according to the experimental conditions explained, that the support presents an absorptive capacity of 2.53 g of laccase per g of support—since at 0.150 g of support, 0.38 g of protein were adsorbed. Moreover, the activity of the support denoted by the new immobilized biocatalyst with a lower ionic strength was 2123 U.kg^−1^ against 1854 U.kg^−1^ under an ionic strength of 0.1 mol.L^−1^ (Table 1), thereby corroborating the increase of recovery activity presented by this support. 

Our promising results can be explained by taking into account the previously performed physical–chemical characterizations of CC support before immobilization (Table 2), together with the scanning electron microscopy (SEM) analysis of the support before and after the degreasing process and the morphological analysis by BET (Figure 5).

By analyzing the C, N, H, and S contents, as well as the determination of cellullose, hemicellulose, and lignin content, we could confirm the lignocellulosic nature of the CC support, which is in accordance with those already reported in the literature [53,58,64]. The higher amount of carbon than other elements, in addition to the considerable percentage of lignin, denoted a high hydrophobic surface; this could be confirmed through the determination of hydrophobicity by BET (Figure 5), where a value of 16.5 μg.g^−1^ was obtained, which characterizes this material as highly hydrophobic, and making it a suitable matrix for the hydrophobic adsorption of L_Asp_. By analyzing the micrographs of the CC support (Figure 5), it is possible to observe that the support in natura is irregular and fibrous and arranged in longitudinal bundles (Figure 5A). After the degreasing process, it is possible to observe intense modification on the surface of the support, which is seen through the formation of pores and culminates in the generation of new forms of interactions between the enzyme and the support, such as accommodation and crosslinking—which are crucial for the acceptance of larger amounts of protein without forming tumultuous layers along the support (Figure 5B).

Besides the hydrophobic surface, the surface area, besides the characterization of the pores of the support, are also factors that need to be investigated during the immobilization process. The BET analysis showed that the surface area of the CC support after the degreasing process was 0.84 m^2^.g^−1^ (all isotherms available in Appendix A). As it is a parameter that reveals the real area available for interaction with the enzyme, its determination is of great importance [65]. 

The volume and pore diameter are also important parameters for the characterization of a good support for enzyme immobilization, since they determine the enzyme load that can be accommodated inside, guiding the immobilization process by a mixture of interactions throughout the support or only through interactions by the surface. Once the support presents a large distribution of pores and the area compatible with the enzyme, the possibility of retaining larger quantities of enzyme in a harmonious way are greater and contribute to a greater retention of activity [51]. According to Tommes et al., 2015 [66], because CC presents a pore diameter of 18.3 Å, this support can be classified as microporous. Porous supports are considered quite versatile due to the fact that they house high protein loads; influence the final properties of the biocatalyst, such as activity, stability, and selectivity; and play an important role in the mass transfer and diffusion of substrates and products. The L_Asp_ applied in this study is a commercial preparation composed by recombinant laccase from *Myceliophthora thermophila* expressed in *Aspergillus orizae*. According to Ernst et al., 2018 [67], the structure of L_Asp_ is a globular, condensed protein, whose dimensions are 70 × 20 × 3 Å, which is compatible for microporous materials and corroborate with the high rates of immobilization efficiency found in this work. 

The CC support presents physicochemical properties a little differently from several other supports and materials that have already been applied for enzyme immobilization. However, it is important to stress that corn cob is a lignocellulosic residue, and as all materials of this class, it presents variations in its content of cellulose, hemicellulose, lignin, and other components, which are influenced by several factors, such as soil, season, harvesting period, and other processing performed. Thus, data on the surface characterization of CC are still scarce in the literature [68,69]. The corn cob needs to be a condensed structure in order to support the corn grains, but it is not the main supporting element of the plant, which is classified as a monocot. For this reason, the lignin contents found in this residual biomass are not as high as in woody vegetables or other biomasses, such as sugarcane. This property is probably also responsible for the small pore volume and its distribution over the surface, as happens with coconut fiber—where the work of Coelho et al., 2012 [54] immobilized laccase on its surface—but this biomass is smooth, with no visible pore size. The corn cob was studied to immobilize laccases by our group because this enzyme is also relatively smaller than other enzymes of industrial appreciation, such as lipases, and given our goal of establishing a simple protocol without many steps of functionalization or pre-treatments. 

Supports from other lignocellulosic sources, such as biochar and sugarcane—among others in the vast majority of available papers—present previous pre-treatment steps, such as alkaline pre-treatment, for the deconstruction of lignin and/or functionalization of cellulose so that covalent bonds could be performed. These protocols often lead to the formation of byproducts that need to be properly managed, as well as adding costs to the process.

### 2.2. Characterization of Immobilized Biocatalyst

For an immobilized biocatalyst to present advantages when compared to the free enzyme, several characterizations need to be performed. An important property of an immobilized biocatalyst is its thermal stability. The association of an enzyme to a support must provide conditions so that the supported enzyme can be protected from hostile conditions, such as temperature increase, which can cause greater vibrations in the protein molecule, thereby leading to thermal denaturation. In this way, both free and immobilised L_Asp_ were submitted to incubation for 1 h at pH 7.0, with 0.05 mol.L^−1^ of phosphate buffer, and at temperatures of 25–65 °C, and their residual activities (difference between enzymatic activity before and after incubation) were determined, whose results are shown in Figure 6.

As can be observed, our immobilized biocatalyst showed greater thermal resistance at higher temperatures when compared to L_Asp_ in its free form. In general, more than 90% of the initial activity present in the immobilized biocatalyst was maintained, with small oscillations until incubation at a temperature of 60 °C, where it presents a relative decrease at temperatures of 65 °C, when it presents around 81% of the residual activity; and 70 °C, when the immobilized biocatalyst demonstrated around 65% of the residual activity (complete values available on Appendix A). These data are quite interesting, since several processes that apply laccases, such as the detoxification of textile dyes, or even organic synthesis, use high temperatures in order to optimize the operation. Therefore, our biocatalyst presents potential resistance to these hostile conditions for free L_Asp_ and can be applied in several processes where this variable is important. Free L_Asp_, as expected and described in the literature, maintained almost all its residual activity at temperatures up to 40 °C, where it is normally considered as its maximum activity temperature. At 45 °C, around 87% of its residual activity was maintained, suffering subsequent decreases until a temperature of 60 °C, where the total inactivation of the enzyme was demonstrated. The immobilization by adsorption allows the deposition of the enzyme along the support—and depending on the shape and composition of the material—entrapment, crosslinking, and other effects can also occur in order to protect the enzyme from hostile environments, such as high temperatures and organic solvents. The increase in temperature causes a greater kinetic effect on the enzyme molecules, which increases their vibration and generates a greater potential for denaturation. Since our support is porous and of high hydrophobicity, the enzyme was probably protected due to the dual action of these characteristics, where part of the enzyme was lodged in the pores of the support; besides being adsorbed with distortion of its conformation, through hydrophobic interactions it can generate a more active and more stable conformation, thereby better resisting the kinetic effects of temperatures above 60 °C than its non-immobilized form. Thus, our biocatalyst demonstrated high thermal resistance, where the mixture of hydrophobic interactions and enzyme accommodation were favorable to the protection of the enzyme against thermal denaturation, and did not allow the immobilisation of L_Asp_ in a more active conformation, which allowed for the expression of high recovered activity in the final biocatalyst.

### 2.3. Application of New Immobilized Biocatalyst on Degradation of RBBR Dye

The new biocatalyst containing the immobilised L_Asp_ under the previously optimised conditions in CC was investigated for the degradation of the textile dye RBBR at pH 7.0 under different temperatures, and was also compared with free laccase. A support control was also performed that aimed to investigate the possible adsorption of the dye. The results are shown in Figure 7.

Since the best results for the immobilization of L_Asp_ were found at pH 7.0, this condition was also applied to evaluate the newly immobilized biocatalyst in the decolorization of anthraquinone RBBR dye. Since RBBR is a very reactive dye, it is characterized as highly soluble in water, which gives stability to the color of fabrics and is able to form covalent bonds between the dye and the fiber [70]. Since it is an anthraquinone, RBBR belongs to the second most used group of dyes, mainly to dye cotton and leather, since it results in strong colors that are resistant to degradation by light, and its chromophore is stable in both acidic and alkaline environments, which can generate both blue, green, and purple shades [71]. Therefore, the choice of pH 7.0 for this experiment does not influence the absorbance in the spectrophotometer of the degradation products [17]. RBBR is a compound derived from anthracene, which is considered a recalcitrant contaminant and has a high toxicity because, although it is highly water-soluble, it is difficult to metabolize by microorganisms because it leads to the formation of reactive oxygen species, which end up generating irreparable damage to DNA [72]. For this reason, the study of new ways of degradation of this dye through enzymes, especially laccases with a robust bioprocess, is of great importance.

As can be seen in Figure 8, the immobilized enzyme showed higher efficiency in the decolorization of RBBR dye, where 64% degradation was observed in 48 h by the new biocatalyst, with improvement in the process in 72 h, when about 74% degradation was obtained. The process was stabilized at 96 h, as can be observed with the corresponding aliquot. It is also noteworthy that from the first aliquots taken, higher degradation values were observed by the immobilized enzyme compared to the free enzyme, which showed degradation results after only 1.5 h of reaction, demonstrating slower kinetics of degradation than its immobilized derivative. These results are different from those obtained by Isanapong et al., 2021 [73], who immobilized the laccase from a *Trametis versicolor* titanium dioxide nanostructure and applied the biocatalyst in the removal of RBBR, thereby obtaining 76% removal in 72 h, compared to about 96% demonstrated by the free enzyme. Moreover, this experiment was performed at pH 5.0, where the laccase from *T. versicolor* showed the highest activity and immobilization yield. Our work has demonstrated, for the first time, the application of an immobilized biocatalyst containing L_Asp_ immobilized in corn cob, a waste product from the agro-industry, where high yields of immobilization and recovered activity were found—as well as high values of RBBR decolorization. Data were not published for L_Asp_ until then, demonstrating that our system has, besides its low cost, promising application in industrial processes, given that the enzyme showed activity until 96 h of investigation. The use of CC as a support for compound removal has also been reported. In the work of Golveia et al., 2021 [69], in natura CC was used for the adsorption of bisphenol-A, a toxic component from the plastic industry. Due to its porous characteristics, in addition to its thermal resistance, the material was able to adsorb up to about 90% of the compound in aqueous solution, making it a promising detoxification agent. Several other supports have been used for the immobilization of laccases with successive application in dye degradation. Hariri et al., 2022 [74] complexed the laccase from *Trametis versicolor* magnetic casein aggregates, with more than 80% removal of crystal violet dye from the medium. Jiang et al. [75] immobilized the same laccase in a one-pot strategy involving the encapsulation of the enzyme in MOF containing Cu, followed by encapsulation in PABA, with subsequent application in the removal of Direct Red 31 dye with about 70% removal in 24 h. Ariaeenejad et al., 2022 [76] synthesized nanocellulose from quinoa agro-industrial waste, which was applied as a support for the immobilization of recombinant laccase (PersiLac1), with efficient removal of two different dyes, malachite green (MG) and congo red (CR), from water. Although there are several supports emerging as good alternatives for the immobilization of laccases, no work thus far has reported on the application of corn cob, a very abundant and resistant agro-industrial waste, as a support for L_Asp_ immobilization. 

One of the greatest advantages of an immobilized biocatalyst is its possibility for recycling, thereby making catalytic systems more efficient. Thus, the newly immobilized biocatalyst was investigated for its capacity for recycling in the decolorization of RBBR dye for 72-h cycles at 35 °C. In this experiment, two different masses of immobilized enzyme were investigated, one regarding the capacity of recycling and the other regarding the maintenance of residual activity (Figure 8).

As observed, the new biocatalyst was able to present, in the first three cycles, similar decolorization efficiencies of the RBBR dye, where it was also possible to observe the maintenance of the residual activity of the support, thereby demonstrating its high operational efficiency—since each reaction cycle was investigated over a total time of 72 h. As observed during the fourth cycle, a direct relationship between the decrease in the residual activity of the support and the reduction in efficiency in the decolorization process can be noted, where the enzymatic inactivation was decisive until the end of the cycles studied. Another aspect that deserves to be highlighted is the fact that the degradation of several textile dyes is also the result of a mutual process between the adsorption of the dye on the support and the action of the immobilized laccase [70,71]. Coelho et al., 2011 [54] studied the immobilization of commercial Novozym 51003 on coconut fibre and the immobilized biocatalyst was applied in the decolorization of various textile dyes. As a result, in the first few cycles, relevant percentages of adsorption of the dyes on the support surface were detected, followed by an increasing participation of the laccase in the degradation of the dyes applied in the study. As shown in Figure 8, a control containing only defatted CC and RBBR at the same study concentration was performed in order to investigate whether the adsorption effect of the dye occurs in our biocatalyst in order to explain the values obtained, since in this case, L_Asp_ was immobilized on a microporous support. As a result, about 10% of the dye is adsorbed on the support (it is important to emphasize that the degradation values demonstrated by the immobilized enzyme were properly discounted from the values found in the control, as well as the study performed with the free enzyme, where only the RBBR solution was used as a control in order to observe some possible degradation with time) during the first 8 h of assay, demonstrating a progressive saturation of the dye on the support. Thus, as exposed above, this saturation also contributed to a greater contribution of laccase activity in a progressive way, since during all cycles, the same portion of immobilized laccase was applied. According to the results of the immobilization efficiency, it becomes more evident that the immobilized laccase was losing its activity throughout the cycles, which was probably accelerated by the excess of dye in the support [71]. As shown in the materials and methods section, the L_Asp_ enzyme that was applied in this work is a commercial mixture containing *Aspergillus* laccase and a mediator, and for this reason, the addition of another mediating compound was not studied and the redox potential of the enzyme was not really investigated. However, our results are different from some works found in the literature, where the operational efficiency of free laccase is still higher than the immobilized derivative, making our biocatalyst interesting for industrial application.

## 3. Materials and Methods

### 3.1. Materials 

Laccase from *Aspergillus* spp. (L_Asp_) with a nominal activity of 1000 U.g^−1^ of protein, 2,2′-azinobis(3-ethylbenzothiazoline 6-sulfonic acid) (ABTS, ≥98%), Bovine serum albumin (BSA), ethyl alcohol anhydrous, orthophosphoric acid 85%, and Coomassie Brilliant Blue G-250 dye were purchased from Sigma-Aldrich (Rio de Janeiro, Brazil). L_Asp_ is a similar product to Novozym 51003, which is a formulation used for indigo dye decolorization in denim finishing operations and includes a buffer and an enzyme mediator, but this information was not indicated by Novozymes [54]. Lignocellulosic waste from corn cob (CC), which was used as a support, were collected in supply and processing centers in the city of Imperatriz, State of Maranhão (Brazil). All reagents used were of analytical grade.

### 3.2. Production of Biocatalyst

#### 3.2.1. Preparation of Support

Initially, the residual biomass of corn cob (CC) was washed and deffated for laccase immobilization following a protocol previously published by our group [55]: 50 g of residual CC was washed 3 times with 100 mL of distilled water and dried for 24 h in a greenhouse (Figure 9A). After this step, the dried material was crushed, sieved (28–35 mesh), and then washed 3 times with 100 mL of ethanol (99.8% PA) at room temperature (Figure 9B). Approximately 10 g of washed support was subjected to degreasing in a Soxhlet extractor for 4 h with 200 mL of 50 °C. After extraction, the sample present in the extraction cartridge was removed and dried in an oven at 60 °C for 6 h (Figure 9C). The final defatted CC was used in the immobilization protocols.

#### 3.2.2. Immobilization Procedures

In this work, the immobilization of the L_asp_ on CC support was carried out by an adsorption protocol. For this purpose, 0.150 g of CC was added to 5 mL of n-hexane and stirred at room temperature for 2 h in conic flasks of 50 mL, followed by filtration and drying overnight [77]. Then, 5 mL of laccase solution (containing 0.001 to 0.5 g.mL^−1^) in sodium phosphate buffer solution with pH 7.0 (0.1 M) was added to the medium under stirring at 4 °C during 24 h at 100 rpm under a shaker. With the aim being to optimize the system, the pH of the immobilization (from 4.0 to 9.0) was investigated with the following buffers at 0.1 M: sodium acetate (pH 4.0–5.0), sodium phosphate (pH 6.0–7.0), and Tris-HCl (pH 8.0–9.0). The immobilization time (15 min to 12 h) and ionic strength (0.05 to 0.2 mol.L^−1^ of phosphate buffer solution at pH 7.0) were also investigated. 

Aliquots of the supernatant were taken in order to quantify the immobilization efficiency through the concentration of proteins and residual activity. At the end of the immobilization processes, the resulting biocatalysts were filtered under vacuum and subsequently washed 3 times with 50 mL of appropriate buffer and dried at room temperature, being subsequently investigated in terms of laccase activity. 

#### 3.2.3. Determination of Laccase Activity

Free and immobilized laccase activities were assayed spectrophotometrically (Model Bel V-M5 Visible NS BE210694) by monitoring the oxidation of 2,2-azino-bis-(3-ethylbenzthiazoline-6-sulfonic acid) (ABTS) substrate (0.4 mM) in 0.1 mM citrate/0.2 mM phosphate buffer at pH 4.5 and 25 °C (coloration changing according to Figure 10).

For measurements, 10 µL of enzyme solution or 10 mg of immobilized enzyme were added to 2 mL of ABTS solution, reaching a final volume of 3.0 mL with the previously cited buffer. The change in absorbance at 420 nm (ε = 36,000 L/mol × cm) was recorded by the UV–vis spectrophotometer every 30 s for 5 min. One unit (U) was defined as the amount of enzyme that oxidized 1 µmol of ABTS per min, and all equations were based on Girelli and Scuto, 2021 [77].

The immobilized enzyme activity (*I_A_*) was determined using the following equation:(1)$$I_{A}\;\left(\frac{U}{kg}\right)=\frac{\Delta abs\times V\times 10^{6}}{\varepsilon \times m}$$
where *I_A_* is the calculated immobilized activity (U.kg^−1^), Δ*_abs_* is the change in absorbance in the corresponding time (min), *V* is the volume of reaction (L), 10^6^ is the conversion factor from M to μM, ε is the molar extinction coefficient of radical cation ABTS^+^ at 420 nm (36,000 L.mol^−1^ × cm), and *m* is the mass of the biocatalyst (kg). The activity yield (*A_i_*), immobilization efficiency (*E_f_*), and recovered activity (*Rec*) on the immobilization processes were also calculated. 

*A_y_* (%) was defined as the percentage of enzyme activity immobilized, taking into account the initial activity of the initial free enzyme. It was calculated according to Equation (2):(2)$$A_{y}\left(\%\right)=\frac{A_{i}-A_{f}}{A_{imob}}\;\times \;100$$

*E_f_* (%) was defined as the percentage of immobilized enzyme in the function of the determined residual activity on the supernatant and was calculated according to Equation (3):(3)$$E_{f}\;\left(\%\right)=\frac{A_{imob}}{A_{i}-A_{f}}\times 100\;$$

*Rec* (%) was determined as the immobilized activity when compared to the initial enzyme activity offered to the system and was calculated according to Equation (4):(4)$$Rec\;\left(\%\right)=\frac{A_{imob}}{A_{i}}\times 100$$

For Equations (2)–(4), *A_i_* and *A_f_* were defined as the laccase activity on the enzyme solution before and after the immobilization protocols, respectively. *A_imob_* was defined as the activity expressed by immobilized biocatalyst. 

#### 3.2.4. Determination of Protein Concentration

The concentration of proteins from the enzyme extract of free L_asp_ and supernatants during the immobilization process was carried out according to the method of Bradford, where the protein content was estimated by the average of a calibration curve obtained using albumin bovine serum (BSA) as a standard [78].

#### 3.2.5. Support Desorption

The new biocatalysts were subjected to protein desorption assays according to the methodology previously described [56,79]: 1 g of immobilized biocatalyst was poured into 50 mL of 25 mM phosphate buffer with pH 7.0 and containing 2.5% (v.v^−1^) of Triton X-100 or NaCl 0.8% (w.v^−1^). The supports immersed in the desorption solution were kept under constant stirring at 100 rpm for 4 h at room temperature. At the end of this period, the particles were separated from the supernatant (vacuum filtered), the supernatant was then subjected to a protein quantification assay and the particulate material was applied in the hexyl laurate esterification reaction.

#### 3.2.6. Thermal Stability

The thermal stability of free and immobilized L_Asp_ in the function of the residual activity was measured as follows: 10 mg of biocatalyst or 10 uL of free laccase was added to 1 mL of phosphate buffer with pH 7.0 (0.05 mol.L^−1^) in cryotubes under stirring in a shaker at 120 rpm for 1 h at temperatures ranging from 25–70 °C. After this procedure, the free and immobilized enzyme were submitted to laccase activity as previously described.

#### 3.2.7. Textural Characterization (BET)

The textural characterization of corn cob after degreasing was determined by porosimetry in a Quantachrome^®^ porosimeter (NOVA-1200), which aimed to evaluate the capacity of adsorption and subsequent desorption of nitrogen at 77 K of the material, thereby forming isotherms of adsorption and desorption. The degassing of the sample under vacuum was performed at a constant temperature of 150 °C for 2 h to eliminate possible contaminants present. From the adsorption isotherm data, the surface area, total volume, and average pore diameter of the material could be determined (Isotherms available on Appendix A) according to the method devised by Brunauer, Emmet, and Teller (BET) [80]. 

#### 3.2.8. Determination of Hydrophobicity of the Support

The relative hydrophobicity of defatted CC was determined by adsorption of the Rose Bengal dye according to the procedure described by Lima et al., 2015 [81]: 0.150 g of CC was added into Erlenmeyer flasks of 125 mL containing 20 mL of a dye solution at 20 μg.mL^−1^. The flasks were kept under stirring for 1 h at room temperature. The samples were filtered, and the supernatant was used to quantify the concentration of the dye through the difference between the initial and final absorbances at 549 nm, and by comparing them with the respective standard curve of the dye. The adsorption efficiency (E) was calculated as the amount of Rose Bengal dye adsorbed per unit area of the supports, which is described by Equation (5):(5)$$E\;\left(m\;.{\;m}^{2}\right)=\frac{\left(C_{initial}\times V_{sol\;\;}\right)-\left(C_{final\;}\times V_{sol}\right)}{m_{support}}÷SA$$
where *C_initial_* is the initial dye concentration, *V_sol_* the volume of the dye solution, *C_final_* is the dye concentration in the final supernatant solution, *m_support_* the mass of the support, and *SA* is the surface area of the support determined by the BET method.

#### 3.2.9. Application on Decolorization of Remazol Brilliant Blue R (RBBR) Dye

The efficiency of new laccase-immobilized biocatalyst on decolorization was carried out using a solution of 50 mg.L^−1^ of RBBR at pH 7.0. For this purpose, 25 mL of each solution was accommodated in conic flasks of 125 mL with 0.2 g of homemade immobilized laccase. The reactions were performed at 35 °C under 120 rpm of stirring. Dye decolorization was determined by monitoring the decrease in the absorbance peak at 580 nm using a Bel V-M5 Visible NS BE210694 spectrophotometer. The comparison of the free laccase and immobilized laccase in RBBR removal was performed by using the same amount of enzyme (U). The defatted CC was applied as a control for the immobilized enzyme; as well, the medium without laccase was applied as the control for the process with free enzyme. Aliquots were taken at times varying between 0.5 and 96 h. The decolorization efficiency (*D_E_*) was determined according to Equation (6):(6)$$D_{E\;}\left(\%\right)=\frac{A_{i}-A_{f}}{A_{i}}\times 100$$
where *A_i_* is the initial absorbance of the RBBR solution and *A_f_* is the final absorbance at a specific time of investigation. 

## 4. Conclusions

In the present work, corn cob residual biomass proved to be a cheap, efficient, and promising support for L_Asp_ immobilization through simple hydrophobic adsorption protocols, where in only 2.5 h, at pH 7.0 and room temperature, 74.8% of the immobilization yield of a 0.1 g.mL^−1^ solution was obtained—with a high recovered activity (39.6%) that generated a biocatalyst with 2123 U.kg^−1^, which are essential parameters in the construction of an immobilized biocatalyst.

BET and hydrophobicity studies demonstrated that defatted corn cob is a microporous and highly hydrophobic material that is ideal for the immobilization of small enzymes, such as laccases, thereby allowing for dual actions of adsorption and accommodation in the pores, which was important for the final biocatalyst to present interesting thermal stability at temperatures up to 70 °C when compared to L_Asp_ in its free form.

In the RBBR decolorization tests, the new biocatalyst showed an operational efficiency during three cycles, and a decolorization rate higher than that of the free enzyme, making it a competitive biocatalyst that is susceptible to industrial applications. This is the first work in which L_Asp_ is immobilized on corn cobs with promising results, as well as application in the degradation of this textile dye, thereby opening precedents for new biotechnological applications.

## Figures and Tables

**Figure 1 ijms-23-09363-f001:**
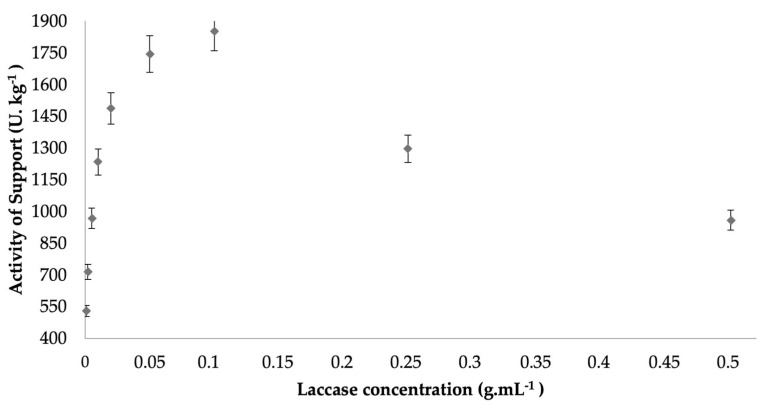
Influence of initial enzyme concentration on the activity of the immobilized L_Asp_ CC support by hydrophobic adsorption at pH 7.0 and 24 h at room temperature.

**Figure 2 ijms-23-09363-f002:**
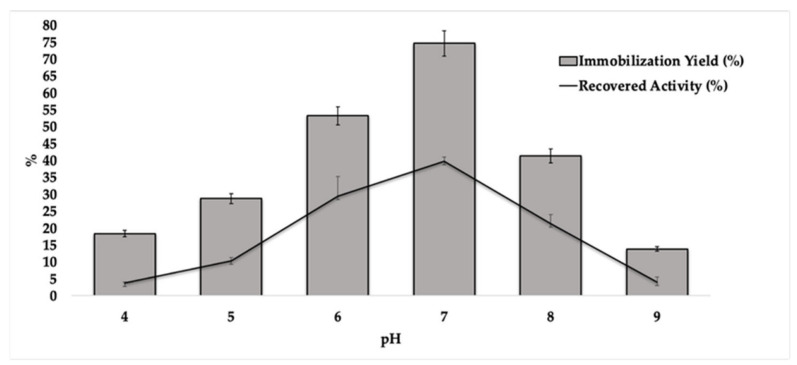
Effect on immobilization yield and recovered activity on immobilization of L_Asp_ on CC support. Experimental conditions: 0.150 g of CC, 5 mL of laccase solution (0.1 g.mL^−1^) at pH 4.0–9.0 (0.1 mol.L^−1^), and 4 °C at 100 rpm under a shaker.

**Figure 3 ijms-23-09363-f003:**
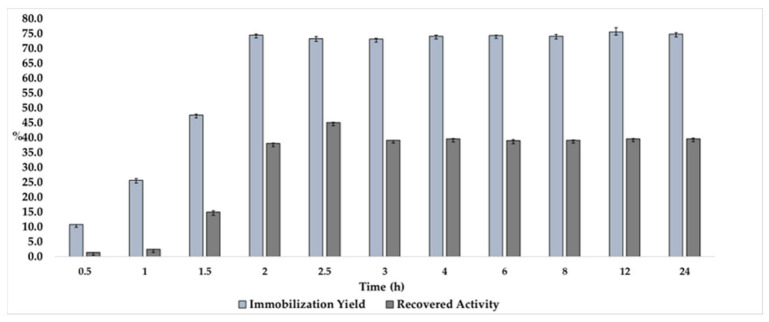
Effect of the time on immobilization yield and recovered activity in the immobilization of L_Asp_ on CC support. Experimental conditions: 0.150 g of CC, 5 mL of laccase solution (0.1 g.mL^−1^) in sodium phosphate buffer solution at pH 7.0 (0.1 M), and 4 °C at 100 rpm under a shaker.

**Figure 4 ijms-23-09363-f004:**
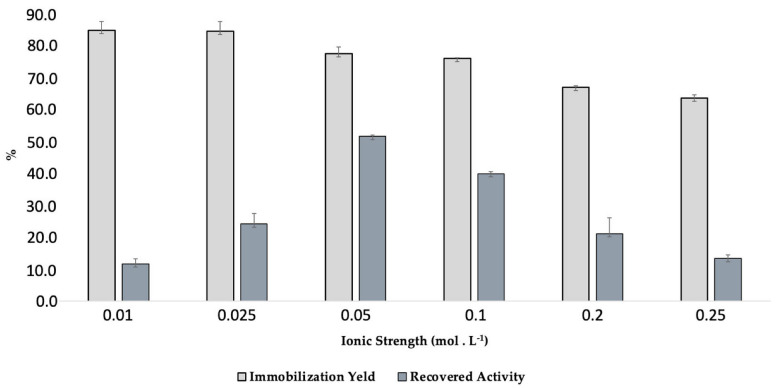
Effect of ionic strength on immobilization yield and recovery activity at immobilization of L_Asp_ on CC support. Experimental conditions: 0.150 g of CC, 5 mL of laccase solution (0.1 g.mL^−1^) in sodium phosphate buffer solution with pH 7.0 (ionic strength in concentration between 0.01–0.25 mol.L^−1^), and 4 °C at 100 rpm under shaker.

**Figure 5 ijms-23-09363-f005:**
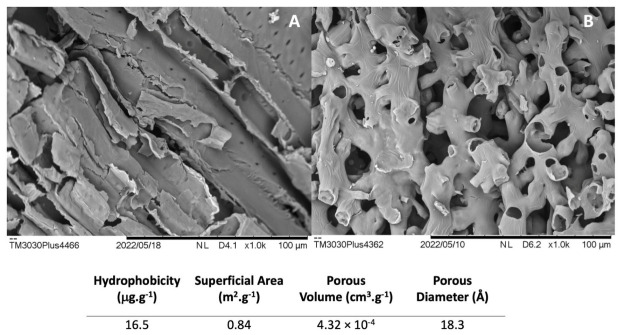
SEM of CC support applied on immobilization of L_Asp_ and hydrophobicity and morphological characteristics measured by BET method and hydrophobicity of CC after degreasing process. (**A**) Corn cob in natura; (**B**) corn cob after degreasing process.

**Figure 6 ijms-23-09363-f006:**
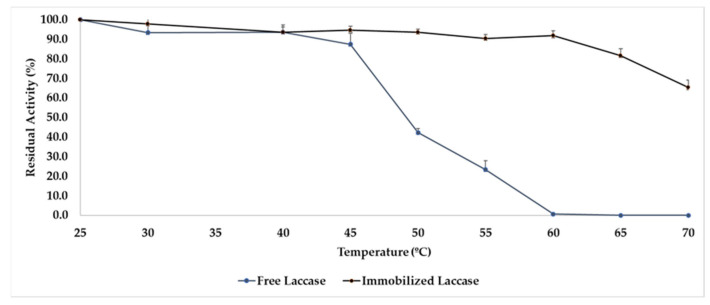
Thermal stability of free and immobilized L_Asp_ in function of residual activity. Conditions: 10 mg of biocatalyst or 10 uL of free laccase in 1 mL of phosphate buffer at pH 7.0 (0.05 mol.L^−1^) in cryotubes under stirring in a shaker at 120 rpm in temperatures ranging from 25–70 °C.

**Figure 7 ijms-23-09363-f007:**
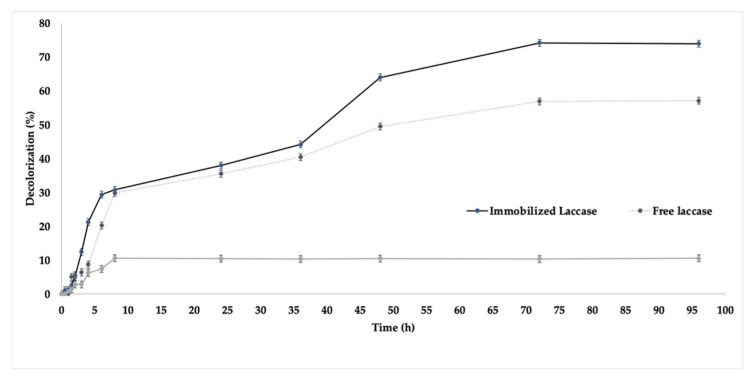
Investigation of decolorization of RBBR catalyzed by free and immobilized L_Asp_.

**Figure 8 ijms-23-09363-f008:**
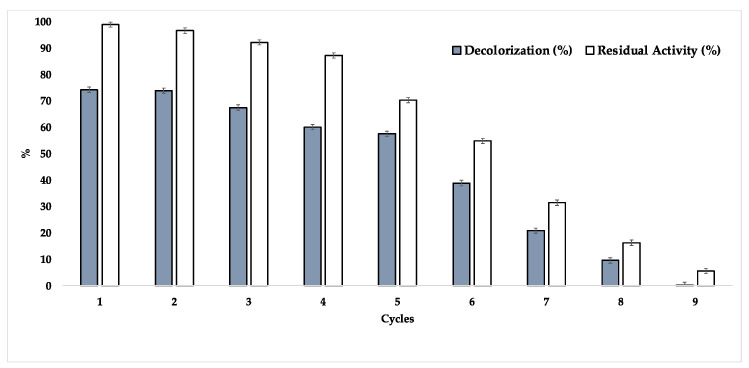
Recycling and determination of residual activity presented by immobilised L_Asp_ in the decolorization of the RBBR dye.

**Figure 9 ijms-23-09363-f009:**
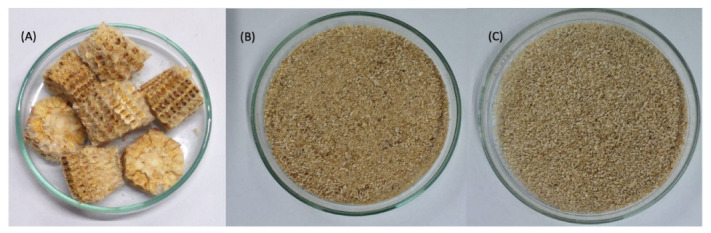
Residual Biomass from corn cob (CC) used as a support for Laccase from *Aspergillus* spp. (L_asp_). (**A**) residual biomass in natura, (**B**) washed and sieved support, (**C**) defatted support. Adapted from [54].

**Figure 10 ijms-23-09363-f010:**
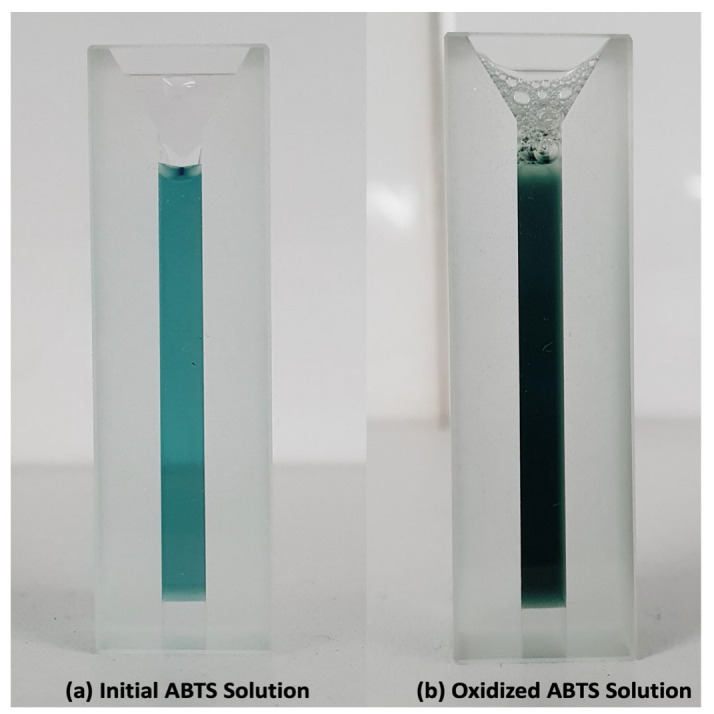
Color changing demonstrated by oxidation of ABTS catalyzed by L_Asp_. (**a**) Initial ABTS solution; (**b**) oxidized ABTS solution.

**Table 1 ijms-23-09363-t001:** Influence of initial enzyme concentration on immobilization yield and recovered activity.

Enzyme Concentration (g.mL^−1^)	Immobilization Yield (%)	Recovered Activity (%)	Activity of Support (U.kg^−1^)
0.001	11.7 ± 5	1.4 ± 4	529 ± 4
0.002	23.4 ± 2	3.4 ± 3	715.2 ± 3
0.005	45.7 ± 1	5.2 ± 2	967.6 ± 2
0.01	57.4 ± 2	6.4 ± 1	1234.7 ± 2
0.02	58.9 ± 6	4.5 ± 7	1487.3 ± 3
0.05	65.1 ± 2	7.7 ± 6	1744.6 ± 5
0.1	74.8 ± 3	39.6 ± 1	1854 ± 1
0.2	38.7 ± 2	16.7 ± 4	1300 ± 2
0.5	21.8 ± 1	16.8 ± 3	959.3 ± 1

Experimental conditions: 0.150 g of support, 5 mL of enzyme solution containing different protein concentrations at pH 7.0 (phosphate buffer 0.1 M), and 24 h of immobilization time at room temperature.

**Table 2 ijms-23-09363-t002:** Chemical composition (%) of CC after degreasing process.

C	N	H	S
47.8 ± 0.1	3.14 ± 0.3	6.13 ± 0.6	˂1
**Cellulose**	**Hemicellulose**	**Lignin**	**Ash (%)**
37.1 ± 0.4	32.1 ± 0.4	9.97 ± 0.5	0.57 ± 0.1

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
