# Peer review of "Corn Cob as a Green Support for Laccase Immobilization—Application on Decolorization of Remazol Brilliant Blue R"

_ijms, 2022, doi:10.3390/ijms23169363_

Round 1

Reviewer 1 Report

Molinares and co-workers investigated the immobilization of a laccase from Aspergillus spp (LAsp), in residual corn cob and evaluated its application in the degradation of Remazol Brilliant Blue R (RBBR) dye. The organization of the manuscript is simple and linear, however I have some comments about their results.

-       - The authors defined as the best conditions of laccase immobilization the one obtained at 0.1 g/mL of enzyme concentrations. In my opinion, comparing results at 0.05 and 0.1 g/mL, the increase of both immobilization Yield and Activity of Support is not enough to justify the use of a double amount of enzyme. Please add a comment.

-      -  Buffer sodium phosphate does not work in the full range of analysed pHs (4-9): how did the authors perform the experiment at pH 4 and 9?

-       - A negative control with not functionalized CC for RBBR decolorization has to be performed and added to the graph (Figure 8), also considering the author’s comment about the effect of the support (pag 13, line 467-480).

-       - Considering the ability of laccases to act on lignocellulosic biomasses, the stability of immobilized laccase on CC during the time should be evaluated, as well as its functionality in RBBR decolorization.

-       - It is not clear if the residual activity (Figure 9) has been measured in the presence of RBBR or not: results have different impact on the study.

 Minor point

-       -Please define the abbreviation for BET (pag 8, line 298)

-       - Please clarify the meaning of the sentence at pag 13, line 482-483 about laccase mediator.

-       -There are many typos throughout the manuscript. Moreover, the organism name should be in italic (Page 12, line 442 and 445)

Author Response

Molinares and co-workers investigated the immobilization of a laccase from Aspergillus spp (LAsp), in residual corn cob and evaluated its application in the degradation of Remazol Brilliant Blue R (RBBR) dye. The organization of the manuscript is simple and linear, however I have some comments about their results.

  - The authors defined as the best conditions of laccase immobilization the one obtained at 0.1 g/mL of enzyme concentrations. In my opinion, comparing results at 0.05 and 0.1 g/mL, the increase of both immobilization Yield and Activity of Support is not enough to justify the use of a double amount of enzyme. Please add a comment.

Answer: Dear referee, thank you for your comment. Our intention in this step of the manuscript was to find a condition where the immobilized enzyme presented a high recovered activity, which demonstrates that the load of immobilized enzyme, presented activity on the support. This parameter is very important to understand which is the best initial enzyme load that should be offered to the support, so that we have the highest amount of enzyme possible with the highest recovered activity. If we compare the data of recovered activity when 0.05mg/mL of enzyme is added (recovered activity 7.7 ± 6 %) and 0.1 mg/mL (recovered activity 39.6 ± 1%), it can be observed that there was an increase of more than 5 times in the recovered activity, with twice the amount of immobilized enzyme, demonstrating that in this system, all the immobilized enzyme was available to access the substrate and exert the ABTS oxidation reaction, which was the standard activity chosen in this study. For this reason, we chose the concentration of 0.1 mg/mL of enzyme to continue the studies.

 -  Buffer sodium phosphate does not work in the full range of analysed pHs (4-9): how did the authors perform the experiment at pH 4 and 9?

Answer: Dear referee, thank you for your comment. In this step of the work, we investigated the effect of pH on the immobilization of laccase from Aspergillus sp. on the defatted support. For this, we performed different buffers that ensured the pH to be studied. Thus, the use of the buffers was as follows:

pHs 4-5: sodium acetate buffer 0.1M

pHs 6-7: Sodium phosphate buffer 0.1M

pHs 8-9: Tris-HCl buffer 0.1M

We also complement this information in the section of materials and methods. This procedure is in accordance to the reference [75] of our manuscript, available at the link https://doi.org/10.1016/j.bcab.2021.102186

- A negative control with not functionalized CC for RBBR decolorization has to be performed and added to the graph (Figure 8), also considering the author’s comment about the effect of the support (pag 13, line 467-480).

Answer: Dear Referee, thank you for your suggestion. We inserted the control of the support in Figure 8, performed with defatted corn cob without the enzyme, and furthermore, performed the recalculation of the values of RBBR dye degradation catalyzed by our immobilized biocatalyst, discounting the value obtained by the adsorption of the dye on the support. It was also done on calculation of recycles, in figure 9. Based on this, we changed a part of our discussion, in order to better adjust this section. Indeed, there was a small adsorption of the dye on the support, which corroborates our discussion raised in this manuscript, in addition to the information published in previous articles.

- Considering the ability of laccases to act on lignocellulosic biomasses, the stability of immobilized laccase on CC during the time should be evaluated, as well as its functionality in RBBR decolorization.

Answer: Dear referee, thank you for your questioning, which is pertinent and interesting. According to the deadline that we have for sending the answers to the referees, there is not enough time to carry out robust studies of stability with time, as well as the maintenance of the activity of the immobilized enzymatic preparation. However, until the moment, the biocatalyst produced in this work is being submitted to other oxidation reactions for future works of our research group, maintaining its catalytic activity. We believe that for LAsp to act properly on the support, it should be subjected to suitable temperatures, in addition to the amount of water and other factors that provide this possible degradation.

-   It is not clear if the residual activity (Figure 9) has been measured in the presence of RBBR or not: results have different impact on the study.

Answer: Dear referee, thank you for your questioning. Yes, in figure 9 there is the graph referring to the recycles performed with the immobilised LAsp and the residual activity study after each cycle. For this reason, we could realize that in this experiment, the immobilized preparation maintained similar residual activities in the first three recycles, as well as was able to present similar efficiencies in the degradation of RBBR dye.

 Minor point

-   Please define the abbreviation for BET (pag 8, line 298)

Answer: The BET Method, also known as Multimolecular Adsorption Theory is a mathematical theory developed to study the physical adsorption of gas molecules on a solid surface, and is a specific surface area analysis technique for materials, such as corn cob. The BET method has this name because it was developed by researchers Stephen Brunauer, Paul Hugh Emmett and Edward Teller (reference available at https://doi.org/10.1021/ja01269a023).

A sub-item in the section Material and Methods was added in order to better contemplate the characterization of the support before and after the immobilization of LAsp as well as the reference was incorporated in the paper.

-  Please clarify the meaning of the sentence at pag 13, line 482-483 about laccase mediator.

Answer: Dear referee, thank you for your question. At the sentence “Because the LAsp in this work does not present mediator, the redox potential of the enzyme was not actually investigated” We refer to the fact that the Laccase from Aspergillus applied in this work, is an enzymatic preparation that commercially is already composed of enzyme and a mediator, since this is a compound similar to Novozym 51003® commercialized by Novozymes®. For this reason, the addition of a mediator was not necessary for the degradation of RBBR dye catalysed by our new biocatalyst to take place. The work of Coelho et al, 2011 (reference 62 of our work, available at doi: http://10.1016/j.molcatb.2011.04.014 ), states that novozym 51003 is a preparation containing laccase from Aspergillus with a mediator. In addition, the technical data sheet for Lacase from Aspergillus spp. marketed by the company Sigma Aldrich®, from which we obtained the product (available at https://www.sigmaaldrich.com/BR/en/sds/sigma/sae0050), states that this is indeed a synonym of Novozym 51003®, corroborating our study. This sentence has been rephrased in the paper for better understanding.

-    There are many typos throughout the manuscript. Moreover, the organism name should be in italic (Page 12, line 442 and 445)

Answer: Done

Reviewer 2 Report

The study investigated the possiblities of immblizaiton of laccase on corn cob for the application of dye degrading. It is of interest and provided good reference for the future researchers. But the following issues should be addressed before it is reconsidered for publication in the journal. 

1.       Line 80 remove one of the “such as”

2.       Line 121 remove "sugar cane"

3.       Line 255-256 The authors claims that “there was a reduction of 12 times for the maximum protein loading”, is this shown in Figure 4?

4.       Make sure the point in all numbers is correct, eg. Line 305 “16,5”, “0,5”in table 2

5.       It is better to provide the chemical composition of the CC before and after the degreasing process, in which extractives removed should be calculated.

6.       Line 316 remove “besides”

7.       The conclusions should be more specific and in details, which include the most important findings in this work.

8.       When discussing the application of laccase immobilized on corn cob to degrade the dye, it is better to compare corn cob with other support in literature.

9.       Check the spelling of words, eg. “Hidrophobicity” in Figure 5

Author Response

The study investigated the possiblities of immblizaiton of laccase on corn cob for the application of dye degrading. It is of interest and provided good reference for the future researchers. But the following issues should be addressed before it is reconsidered for publication in the journal. 

  1. Line 80 remove one of the “such as”

Answer: Done

  1. Line 121 remove "sugar cane"

Answer: Done

  1. Line 255-256 The authors claims that “there was a reduction of 12 times for the maximum protein loading”, is this shown in Figure 4?

Answer: Dear Referee, thank you for the question. In figure 4 it is shown that in the times from 2 to 24h, similar values of immobilization yields (%) were found. In the previous figures, it was shown that we started the work applying 24h of time to evaluate the best pH and the best amount of laccase loading for the continuation of the study. Considering that similar values were found both in 24 and 2h of process, we conclude that there was a reduction of 12h in the immobilization time, where the maximum of protein was adsorbed in the system. 

  1. Make sure the point in all numbers is correct, eg. Line 305 “16,5”, “0,5”in table 2

Answer: corrected

  1. It is better to provide the chemical composition of the CC before and after the degreasing process, in which extractives removed should be calculated.

Answer: Dear referee, thank you for your question. In this work, we initially performed an experiment in an attempt to promote the immobilization of LAsp in corn cob just crushed, washed and sieved. However, as the results obtained were low (21% in 72h), even at pH 7, we decided to perform the degreasing process of the residual biomass, following previous works of our research group, which denoted that other residual lignocellulosic biomass were more promising as a support for the immobilization of enzymes after the degreasing process. Therefore, we only performed the whole physicochemical characterization of the defatted material. The short deadline for delivering the review of this manuscript also prevents us from performing this further study.

  1. Line 316 remove “besides”

Answer: corrected

  1. The conclusions should be more specific and in details, which include the most important findings in this work.

Answer: Dear referee, we have completed the conclusion section.

  1. When discussing the application of laccase immobilized on corn cob to degrade the dye, it is better to compare corn cob with other support in literature.

Answer: Dear referee, thank you for your suggestion. We completed this discussion part on the paper.

  1. Check the spelling of words, eg. “Hidrophobicity” in Figure 5

Answer: corrected

Reviewer 3 Report

The authors immobilized Laccase over renewable corn cob through adsorption and explored the optimal immobilization conditions. The synthesized biocatalysts showed enhanced thermal stability, higher reactivity, and longer catalyst life towards degradation of RBBR dye. 

After review of the manuscript, my overall recommendation is “Accept after minor revision”

Major questions in the manuscript to be addressed are:

1. Please elaborate the reason to use defatted corn cob. What is the immobilization yield and activity using corn cob in natural state after washed and sieved? 

2. Please clarify the meaning of Eq 2. Since Eq 3 described the immobilization efficiency, which is usually no more than 100%, would Eq 2 always give you a no less than 1 value? What’s the physical meaning behind Ay(%)?

3. BET is part of the physisorption experiments that only determines material surface area. How was hydrophobicity determined? Please provide details on the equipment and experiment conditions. Please also provide adsorption isotherm in the supporting materials. Moreover, I would highly recommend the authors to compare the difference in surface area, pore volume, pore size distribution, etc. between corn cob in nature state and corn cob after degreasing in the tabulated format.

4. An ideal support to immobilize enzymes is one that has moderate surface area and pore size distribution for enzymes to access to its internal surface area. The degreased corn cob showed small surface area of 0.84 m2/g and tight pore diameter of 18.3 Å, which is much smaller than other agro-industrial waste materials such as eggshell, cellulose nanofibers, biochar, to name a few. How does the corn cob compare to these materials as the support? Please add more relevant discussions for the interest of readers. Furthermore, the pore size of corn cob seems to be too small to accommodate majority enzymes that are larger. What would you do to improve in the future to broaden the application? 

5. How much free laccase was used in the degradation of RBBR dye in comparison of immobilized laccase? Was 0.2g mentioned in Section 2.2.6 for immobilized laccase including the weight of support? Please clarify and compare the rate of decolorization under the same amount of laccase to demonstrate the efficiency between the two catalysts. 

6. Why does the decolorization rate increase from 35h to 50h significantly using immobilized laccase? 

Minor questions in the manuscript to be addressed are:

1. Repetitive type of “such as” in Line 80.

2. Please correct the decimal used in the manuscript. In Line 321, please change from 0,84 to 0.84. Same corrections should also be implemented in Table S1.

3. The decolorization% of RBBR is evaluated the same way as any chemical reaction conversions. Is Eq(5) calculated as (Ai-Af)/Ai, not (Ai-Af)/Af?

Author Response

Dear referee, thank you for your suggestions. All answers are attached at the file below.

Reviewer 4 Report

The present research article utilized corn cob as a support material for laccase immobilization and applied it to remove the textile dye (decolorization of dyes from water). However, the support material is economical but very conventional in origin. As many excellent supports are emerging, this study looks conventional to fit in a journal such as IJMS. My specific comments are as;

1.     In the preparation of support, the authors degreased the support materials. The author should provide the data on how this process benefited the immobilization of laccase by comparing it with control non-degreased CCs.

2.     Give space between value and units in all the manuscript.

3.     The authors mixed 0.150 g of CC with n-hexane, stirred for 2 h, and added it to the laccase solution. The significant comment arises as n-hexane is the organic solvent, and laccase is proteinaceous. This solvent might inhibit enzyme activity. The author should provide a possible explanation for this.

4.     Authors expressed activities in U/KG. However, the author used 10 mg of the support for immobilization activity measurement. So, I recommend expressing immobilization data of activity obtained per milligrams of support material. It will give more realistic data. As Mg to Kg is a very high conversion factor. Also, I recommend adding photographic images of the color change in immobilized laccase activity.

5.     Also, the author should give the data for the adsorption capacity of the support material (mg of laccase immobilized / gram of the support) by measuring the initial solution and after immobilization (retained solution) protein concentrations. It gives the material capacity comparable with the recent supports.

6.     Line number 165, page 4, the author mentioned Ai is Activity yield. Then what Ay (%) stands for. The author needs to look at this formula very carefully. Ay expressed in %; however, not multiplied by 100 in formulae. The author should check all the formulae carefully and give valid references for all formulas

7.     The authors should explain how pH 7 gives higher immobilization than another pH. What is the mechanism behind it?

8.     Check the spelling of hydrophobicity in Figure 5.

9.     Figure 6 should be sequenced after Figure 4, as effect pH, time, and buffer ionic strength can be seen sequentially.

10.  The sentence (Line number 351, page number 10) “According to the literature, the best pH for the immobilization of lacases has been pH 7.0.” need to be corrected as, immobilization pH varies on many factors, such as nature of support, type of immobilization, type of modification etc.

11.  Looking at the nature of the immobilization as hydrophobic adsorption, the results obtained in thermal stability (Figure 7) at 65 and 70 oC need more explanation? Also, the authors do not provided the details of the thermal stability experiment in the methodology section.

12.  The authors measured the dye decolorization spectrophotometrically. However, it is well known that dyes can be very easily adsorbed on agricultural wastes. The author should put the control support material without laccase to eliminate the dye removal by adsorption. The obtained results might be due to adsorption as well. As the authors did not monitor degradation via different techniques, authors should use the terminology of decolorization for this section.

13.  Authors got decolorization of dye Remazol Brilliant Blue R by laccase without redox mediator (as I carefully checked the methodology of decolorization experiment, there is no mention of redox mediator addition). As laccase mostly required the redox mediator to carry out degradation of environmental pollutants. If not, the authors should explain the obtained results.

Author Response

(The authors gave the same response as above.)

Round 2

Reviewer 2 Report

After further review the revised manuscript, I recommend to accept it in its current form. 

Author Response

Dear refereem thank you for your comment. We have now made the required editor corrections.

kind regards